# Influence of teaching a structured and humanized method of care on the perception of medical student attitudes in the doctor-patient relationship

**Higor Chagas Cardoso** [1,2] *, **Edna Regina Silva Pereira** [3], **Viviane Soares** [2,4], **Marcelo Fouad Rabahi** [1]

**1** Postgraduate Program in Health Sciences, Federal University of Goias, Goiânia, Goiás, Brazil, **2** Medicine Course, Evangelical University of Goias, Anápolis, Goiás, Brazil, **3** Postgraduate Program in Health Teaching, Federal University of Goias, Goiânia, Goiás, Brazil, **4** Postgraduate Program of Human Movement and Rehabilitation, Evangelical University of Goias, Anápolis, Goiás, Brazil

☯ These authors contributed equally to this work.
* medhigor@gmail.com

**Data Availability Statement:** All relevant data are within the manuscript and its Supporting information files.

## Abstract

### Introduction

The humanization of care can be defined, in a generic way, as the act of making an empathetic and respectful approach to patients. This study proposed to evaluate the perception of attitudes of medical students regarding the doctor-patient relationship, after implementation of teaching a humanized and structured care method.

### Materials and methods

Single-blind, randomized controlled experimental study that evaluated medical students in relation to patient care, based on a pre-post design, using the Patient-Practitioner Orientation Scale (PPOS). This scale has been validated to assess patient-centered attitudes, as the prime outcome measure. The intervention, with a group of randomized students, included teaching the structured and humanized method of patient care, denominated the SEAGULL (Subjective, Exams, Analysis, Goal, Ultimate Action), and was carried out at the university outpatient clinic.

### Results

Fifty-nine medical students participated in the study, with a mean age of 21.3 years (SD = 2.8) and a higher prevalence of female students (71.2%). The increase in the final scores was greater in the intervention group (p = 0.025) when comparing means of the total PPOS scores. The intervention group presented a larger effect size and higher mean scores (d = 0.49, Δ = +0.38, p<0.001) than the control group (d = 0.21, Δ = + 0.10, p = 0.004). It is noteworthy that the analysis of the initial and final means of the PPOS scores of the sharing domain revealed larger effect sizes in the intervention group compared to the control group (Δ = +0.42, d = 0.63; p<0.001).

**Funding:** The author(s) received no specific funding for this work.

## Discussion and conclusion

The findings showed that training in the use of the SEAGULL structured method led to a significant increase in PPOS scores related to the humanization of care by these students, with emphasis on the domain of sharing information, power, and responsibility with patients.

## Introduction

The humanization of care can be defined, in a generic way, as the act of making an empathetic and respectful approach to patients. Its identification and understanding are related to attitudes guided by the perception and appreciation of the subjects, through ethical and humane conduct [1–4].

The pandemic caused by the new coronavirus (Sars-Cov-2) in 2020 was characterized by the imposition of restrictive measures that limited physical contact between patients and their respective assistant health professionals, friends, and family. As a result, there has been an increase in discussions about how to achieve the humanization of care with so many safety barriers. Also of note is the implementation of information and communication technologies, in order to structure care and break the physical measures of isolation [5–8].

Bearing in mind the structuring of care in a humanized view, with the implementation of communication skills and understanding of the expectations of the patient and their families, Rabahi [9] proposed the introduction of the goal of humanized care. The increase in the goal of care, together with the entire assistant health team, enables more humanized care from a more empathic point of view, with the centralization of care on the patient [9].

It is believed that teaching this method of humanized and structured care to medical students could also improve their attitudes towards care regarding the doctor-patient relationship, promoting the development of skills that will qualify the care offered by future doctors when they are in the job market, and enabling them to come closer to finding solutions to the health problems faced by the country's population.

The existing literature on medical education already demonstrates the importance of technical-scientific and biological knowledge for the training of medical professionals. However, it is necessary to expand the discussion on the benefits of teaching elements of humanization policies in the competencies of medical curriculum guidelines. Therefore, studies like the current one are needed to demonstrate how the development of these teaching strategies, that increase students' recognition of patients' expectations and autonomy, can be implemented, in addition to improving the perception of a more humanized and beneficial doctor-patient relationship [10].

Thus, the present study aimed to evaluate the perception of the attitude of medical students regarding the doctor-patient relationship after implementation of the teaching of a humanized and structured care method.

## Materials and methods

This is a single-blind, randomized controlled experimental study, with a quantitative approach. The sample was composed of academics from the fifth academic semester of the medicine course at the Evangelical University of Goiás (Brazil), which uses the Problem Based Learning (PBL) teaching methodology. The research project was approved by the Research Ethics Committee of the institution, under CAAE: 25267019.7.0000.5076 (2020). Recruitment

started on August 3, 2020 and ended on November 30, 2020. The Informed Consent was in written form and was signed by all participants. This study did not include minors.

Considering a significance level of 5%, a sample size equal to 23 subjects for the intervention group and 36 subjects for the control group, and a standard deviation equal to 0.46 mu., from a previous cross-sectional study with this population, a sampling power of 92% was calculated by PSS Health software [11] to test whether there was a minimum difference of 0.42 in the mean PPOS score.

After authorization from the authors, the *Patient-Practitioner Orientation Scale—PPOS* [12] was used, translated and validated for Brazilian Portuguese [13]. The PPOS is a validated scale used to evaluate the attitudes of patients, doctors, and medical students regarding the doctor-patient relationship. The scale is based on items that reflect domains related to the attitudes of "sharing" and "caring" for patients [12].

Items in the "sharing" domain assess whether respondents believe that power and control should be shared between the physician and the patient, and the degree to which the physician should share information with the patient [12–16].

The questions inherent to the "caring" domain demonstrate whether the evaluated participants consider the expectations, feelings, and lifestyle of patients as critical elements of the doctor-patient relationship [12, 14].

The PPOS consists of 18 items, 9 for Sharing and 9 for Caring, using a Likert scale ranging from 1 (strongly agree) to 6 (strongly disagree), based on the calculation of the mean of the scores of all items (total score), for the nine items of each domain ("caring" and "sharing") [12] and for the areas corresponding to the patient-centered communication model [15]. Higher mean scores on the PPOS scale demonstrate patient-centered attitudes. Other authors using the scale reported cut-offs such that a mean of >4.57 reflects a preference for a more patient-centered relationship [12, 16–18].

In addition to the PPOS scale, a sociodemographic questionnaire was applied to analyze the variables age, sex, family income, origin, participation in scholarship programs, extracurricular internships, scientific initiation, artistic activities, the presence of serious personal or family illnesses, parents' education and profession, desired specialty, and religion.

The PPOS scale and the sociodemographic and curricular questionnaire were applied to medical students face-to-face, randomized into two groups. Double blinding was carried out by an assistant professor from the Medical Skills department, in the fifth semester of the Medicine course, without the influence of the researchers. In this subject, students were divided into groups of 10, totaling ten groups (G1-G10). The activities were carried out in a rotating format, between the classroom (50%, or five groups) and the university pulmonology outpatient clinic (remaining groups). The teacher explained the study to the students and that they would be allocated into two groups, but would not know whether they were in the experimental or control group. Although the invitation was extended to the entire class, only 80 students agreed to participate (Fig 1) [19]. A draw was subsequently carried out to determine which groups would be subjected to the experimental protocol.

The control group consisted of the first 36 students who participated in the practical pulmonology rounds at the beginning of the academic semester. This control group did not receive any change in the standard teaching plan provided for in the Pedagogical Project of the Course of medicine.

The intervention group was composed of 23 students who attended the practical rounds of Pulmonology, carried out as part of the teaching plan provided for in the Pedagogical Project of the Course. This group learned knowledge and skills related to the SEAGULL structured and humanized service method, proposed by Rabahi [9].

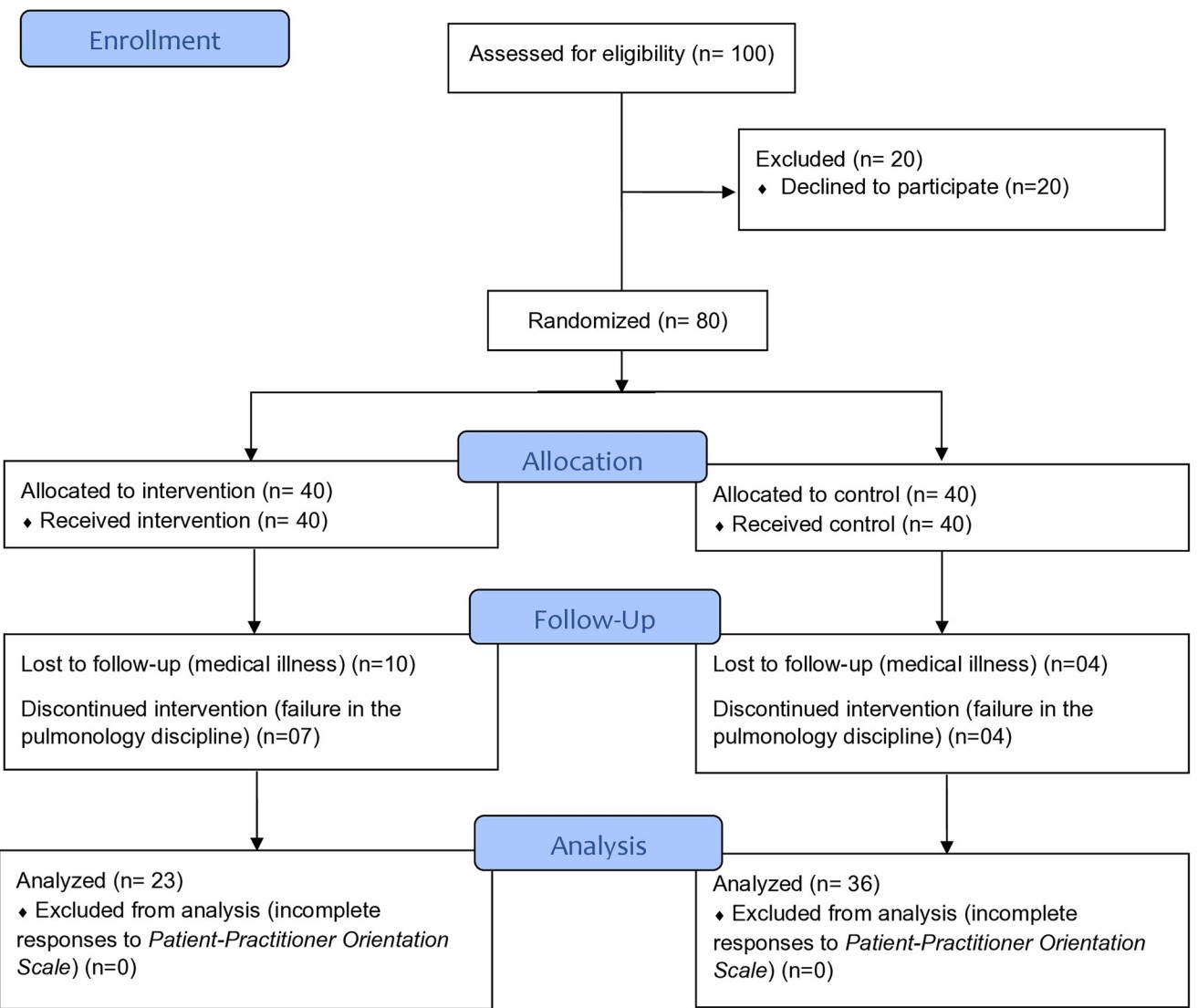

**Fig 1. Consolidated standards of reporting trials flow diagram for participant recruitment and randomization.**

The intervention began in the first practical class of Pulmonology, with the presentation of a theoretical video class developed by the researchers, lasting 20 minutes, which introduced knowledge related to the humanization of care, and humanized communication and care centered on patient expectations. In addition, the video class presented the SEAGULL service method [9] denominated by the acronym of the initial letters of the words Subjective, Exams, Analysis, Goal, Action. This method begins with an anamnesis in search of symptoms and subjective information, during which signs are collected through physical examination and recording of any complementary tests performed. The doctor and/or medical student then proceeds to the analysis of subjective information and exams, after which a goal is planned, to be achieved in the next consultation. Finally, the conduct and actions necessary to achieve the goal of care are determined, together with the patient and their families (Fig 2).

After watching the video class, the intervention group continued to be guided and instructed by professors not belonging to the group of researchers, to apply the SEAGULL care

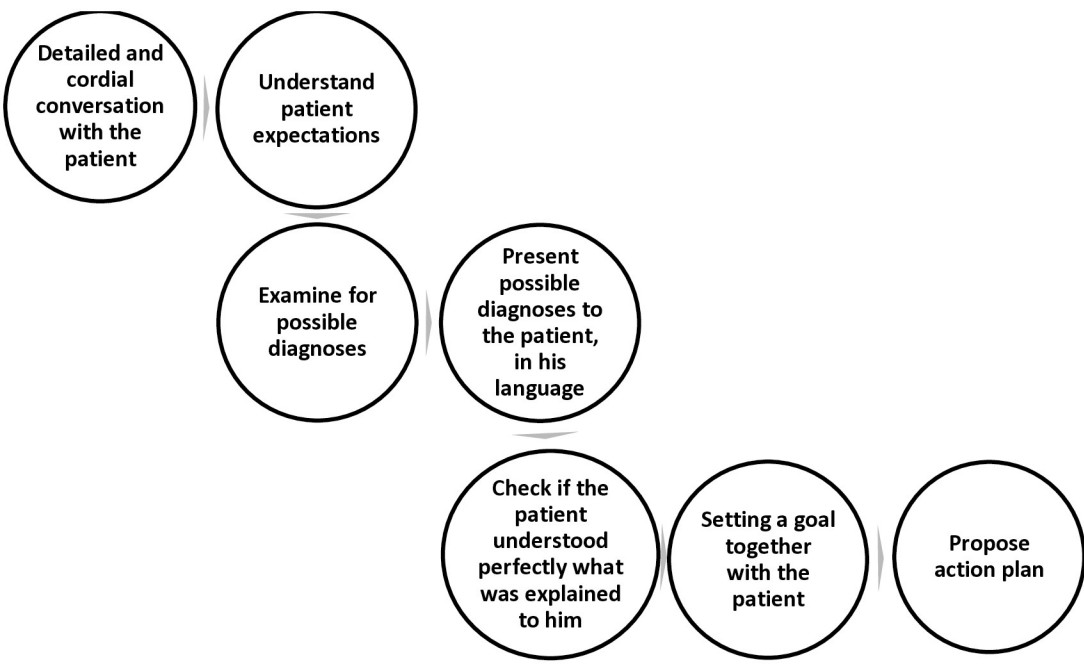

**Fig 2. SEAGULL method.**

method, during eight practical classes of patient care. These activities took place at the university pulmonology outpatient clinic, over an interval of one academic month.

At the end of the semester of medical skills module V, the researchers re-applied the PPOS scale and the sociodemographic and curricular questionnaire to the two groups evaluated.

Data were assessed using the *Statistical Package for Social Science* (*SPSS*) and described as means, standard deviations, frequencies, percentages, and graphs. The association between the categorical variables was verified by the chi-square test and, if necessary, the *Likelihood Ratio* correction was used. To verify the normality of the quantitative variables (discrete and continuous), the Shapiro Wilk test was used. Variables with normal distribution were compared using the Student's t test for paired or unpaired samples and those with asymmetric distribution using the Wilcoxon test.

To verify the interaction between the initial surveys and the final test and to compare PPOS scale measures pre and post-intervention, the variation in the means for each group was used, through the unpaired Student's t-test and Mann Whitney test. In addition, the Cohen's d effect size (ES) was calculated [20]. The value considered significant for p was $\leq 0.05$.

## Results

In the present study, the final sample consisted of 59 medical students, with a mean age of 21.3 years (2.8), a higher prevalence of female students (71.2%), coming from the capital of the state (44%), and Catholic (50.9%). The majority of students reported not participating in student financial aid scholarship programs, extracurricular activities, and arts. However, 84.7% of students reported participating in activities related to scientific initiation (Table 1).

The majority of the sample of students did not report a personal history of serious illnesses (98.3%), but in relation to family history, 59.3% reported these diagnoses. It was also observed

**Table 1. Sociodemographic and curricular characteristics of the 59 medical students.**

| | Intervention (n = 23) | Control (n = 36) | p* |
|---|---|---|---|
| | n (%) | n (%) | |
| *Age (years)* | 20.65 (2.60) | 21.78 (2.86) | 0.35 |
| *Sex* | | | |
| Female | 16 (38.1) | 26 (61.9) | 0.83 |
| Male | 07 (41.2) | 10 (58.8) | |
| *Family income* | | | |
| <10 minimum wages | 12 (46.2) | 14 (53.8) | 0.006 |
| 10–20 minimum wages | 03 (14.3) | 18 (85.7) | |
| 20–40 minimum wages | 07 (77.8) | 02 (22.2) | |
| >40 minimum wages | 01 (33.3) | 02 (66.7) | |
| *Origin* | | | |
| Capital of state | 08 (30.8) | 18 (69.2) | 0.34 |
| Interior of state | 14 (48.3) | 15 (51.7) | |
| Capital city | 01 (25.0) | 03 (75.0) | |
| *Religion* | | | |
| Catholic | 15 (50.0) | 15 (50.0) | 0.11 |
| Evangelical | 07 (35.0) | 13 (65.0) | |
| Spiritualist | 01 (16.7) | 05 (83.3) | |
| None | 0 (0) | 03 (100.0) | |
| *Student scholarship* | | | |
| Yes | 10 (50.0) | 10 (50.0) | 0.21 |
| No | 13 (33.3) | 26 (66.7) | |
| *Extracurricular internship* | | | |
| Yes | 06 (33.3) | 12 (66.7) | 0.56 |
| No | 17 (41.5) | 24 (58.5) | |
| *Artistic activity* | | | |
| Yes | 06 (37.5) | 10 (62.5) | 0.89 |
| No | 17 (39.5) | 26 (60.5) | |
| *Scientific research* | | | |
| Yes | 19 (38.0) | 31 (62.0) | 0.72 |
| No | 04 (44.4) | 05 (55.6) | |
| *Personal illness* | | | |
| Yes | 0 (0) | 01 (100.0) | 0.32 |
| No | 23 (39.7) | 35 (60.3) | |
| *Family Illness* | | | |
| Yes | 14 (40.0) | 21 (60.0) | 0.85 |
| No | 09 (37.5) | 15 (62.5) | |
| *Father's schooling* | | | |
| Fundamental | 05 (71.4) | 02 (28.6) | 0.16 |
| Middle | 05 (31.3) | 11 (68.8) | |
| Higher | 13 (36.1) | 23 (63.9) | |
| *Mother's schooling* | | | |
| Fundamental | 01 (25.0) | 03 (75.0) | 0.43 |
| Middle | 07 (53.8) | 06 (46.2) | |
| Higher | 15 (35.7) | 27 (64.3) | |
| *Medical parent* | | | |

(*Continued*)

**Table 1.** (Continued)

|  | Intervention (n = 23) | Control (n = 36) | p* |
|---|---|---|---|
|  | n (%) | n (%) |  |
| Yes | 01 (30.0) | 04 (80.0) | 0.34 |
| No | 22 (40.7) | 32 (59.3) |  |

* Data for p≤ .05 statistically significant.

that children of parents who attended higher education prevailed in the sample (61%), with 8.4% of the students having parents who had graduated in medicine.

The majority of students reported a family income of less than 10 minimum wages, with a statistically significant difference being described between the family incomes declared by the students in the group that received the intervention and the control group (p = 0.006). However, no positive association was observed (p>0.05) when comparing the domains of PPOS scores and the family income of the sample evaluated.

According to the analysis, the mean of the sum of the PPOS scale scores obtained by the sample, as well as their domains "caring" and "sharing" and their respective areas of communication, showed that at baseline the students demonstrated attitudes centered on the doctor and on the disease (mean scores <4.57) (Table 2).

When considering the caring domain of the PPOS, statistically significant differences were identified (p = 0.042) between the initial and final means of the students evaluated in the intervention group (Δ = +0.30, p = 0.002) and in the control group (Δ = +0.20, p = 0.03).

Analysis of the initial and final means of the PPOS scores in the sharing domain demonstrated a higher effect size in the intervention group than in the control group. In the group that received the intervention, higher values were identified in the total scores of this domain (Δ = +0.42, $d$ = 0.63; p<0.001).

In the analysis of interactions between the groups, the increase in the score at the end was greater in the intervention group (p = 0.025) when compared to the means of the total PPOS scores. The intervention group (Δ = +0.38, p<0.001) presented a greater increase than the

**Table 2. Comparison of PPOS scale measures, pre and post-intervention, and interactions between groups (n = 59).**

|  | Intervention (n = 23) | | | | Control (n = 36) | | | | Interaction between groups |
|---|---|---|---|---|---|---|---|---|---|
|  | Initial | Final | | | Initial | Final | | | |
|  | Mean (SD) | Mean (SD) | p* | ES (*d*) | Mean (SD) | Mean (SD) | p* | ES (*d*) | p* |
|  | Med (Q1–Q3) | Med (Q1–Q3) | | | Med (Q1–Q3) | Med (Q1–Q3) | | | |
| **Caring** | 3.88 (0.92) | 4.18 (0.90) | **0.002** | 0.33 | 4.26 (0.47) | 4.46 (0.61) | **0.03** | 0.37 | **0.042** |
|  | 4.11 (3.67–4.44) | 4.44 (3.89–4.78) | | | 4.28 (4.11–4.56) | 4.56 (4.03–4.89) | | | |
| **Sharing** | 3.67 (0.63) | 4.09 (0.71) | **<0.001** | 0.63 | 3.97 (0.67) | 3.98 (0.57) | 0.97 | 0.27 | 0.09 |
|  | 3.67 (3.22–4.11) | 4.11 (3.78–4.56) | | | 4.06 (3.47–4.39) | 3.89 (3.56–4.44) | | | |
| **PPOS** | 3.78 (0.70) | 4.14 (0.77) | **<0.001** | 0.49 | 4.12 (0.43) | 4.22 (0.53) | **0.004** | 0.21 | **0.025** |
| **Total mean scores** | 3.89 (3.56–4.17) | 4.28 (3.94–4.56) | | | 4.11 (3.85–4.44) | 4.19 (3.79–4.69) | | | |

SD-standard deviation; Med-median; Q1-first quartile; Q3-third quartile; ES- effect size; Cohen's d-d.

*Data for p≤ .05 statistically significant.

Differences pre-post were evaluated by the paired Student's t-test (normal distribution) and Wilcoxon test (asymmetric distribution). Interactions between groups were performed using the comparisons of variation between means pre-post in each group.

control group (Δ = +0.10, p = 0.004) and the ES was also higher in the intervention group (*d* = 0.49) compared to the control group (*d* = 0.21).

## Discussion

The current study demonstrated, through the PPOS scale, that the use of the structured care method with elaboration of goals (SEAGULL) led to an increase in PPOS scores related to the humanization attitudes of medical students, with centralization of care on the patient. The elaboration of a goal to be achieved during humanized care enables the centralization of attention and care in the person of the patient, as well as expanding communication in the doctor-patient relationship, in order to share decision-making [9, 12, 13, 16, 17].

However, despite the increase in PPOS scores, the mean PPOS scores in the current study showed a prevalence of medical students with attitudes centered on the physician and disease (mean scores <4.57), as in previously published studies [20–30]. These results diverge from research carried out in other Brazilian medical schools [16, 31, 32] which demonstrated a prevalence of medical students with mean PPOS scores associated with moderately patient-centered attitudes (mean score 4.57–4.99). This difference may be related to the sample size and the period of the course in which the students were enrolled, since the sample evaluated by the current study was composed of academics from the fifth academic semester of the medicine course, while other national studies analyzed academics from different undergraduate periods. The PPOS scores of students at the end of graduation tend to be positively associated with patient-centered attitudes, as already demonstrated by Ribeiro et al. [32] and Ahmad et al. [21].

The current study identified that the mean scores related to the caring domain of the PPOS scale were higher than the scores of the sharing domain, as observed in other national studies [16, 32]. However, when analyzing the sharing domain, a higher ES and a statistically significant difference (p≤0.05) between the initial scores and the final scores were evidenced only in the group that received the intervention, through the teaching and guidance of the use of the SEAGULL method [9]. It is noteworthy that the group that received the intervention showed a medium ES (0.50–0.79) when evaluating the scores related to the area of communication of sharing with patients, and in the control group this area of communication presented a small ES (0.20–0.49).

Although the effects of using the SEAGULL care method proposed by Rabahi [9] have not yet been evaluated in other studies with medical students, it is suggested that its use was responsible for the differences between the intervention and control groups.

There was a perception that learning and applying the SEAGULL care method allowed students in the intervention group to develop attitudes of empathy and patient-centered communication skills. This method of structured care led to the application of broad medical knowledge in favor of the best treatment for the patient and implementation of the attitudes of academics to center care on the patient. Therefore, it is proposed that the SEAGULL care method also provided these future doctors with the skills needed to develop and structure goals appropriate to humanized care.

It is known that students from medical schools that use active teaching methodologies, such as PBL [16, 33], and those who participate in national humanization programs, such as the Gold Humanism Honor Society [32], have higher PPOS scores. These students are encouraged to develop communication skills that endow academics with more humanized and patient-centered care attitudes. The results of the current study, that introduced the teaching of the SEAGULL method, also showed significant alterations in PPOS scores in a university that uses PBL. An improvement was observed in the students' attitudes related to the areas of

communication for sharing information, power, and responsibilities with the patient and, therefore, implemented the attitudes of humanization of students with centralization of attention on the patient.

The current study presents limitations related to the self-reported data during the completion of the questionnaires and that we did not evaluate qualitative variables, which may have led to underestimation or overestimation with regard to the evaluation of practical attitudes of humanization of care.

In addition, due to the pandemic period in which the project was carried out, during the year 2020, this study had a significant problem with respect to the sample size, with the loss of 24 individuals from the study population because of medical illness. Thus, the final intervention group contained only 23 participants, restricting further evaluations between the group that received the intervention and the control group, as well as making it impossible to continue the investigation with other medical students at the study site.

## Conclusions

Thus, teaching the SEAGULL humanized and structured care method, in a sample of medical students, led to an increase in PPOS scores that evaluated the humanization attitudes of these students regarding the doctor-patient relationship. The positive association of the SEAGULL method with the implementation of PPOS scores in the areas of communication for sharing information, power, and responsibilities with the patient is highlighted.

The results of the current study, which evaluated humanization in medical education, suggest the need for future research that demonstrates the importance of training programs for medical students, in order to reverse the current panorama of medical care, focused on attitudes centered on the physician and disease.

## Supporting information

**S1 File.**
(DOCX)

## Acknowledgments

We would like to thank Professor Edward Krupat (Harvard Medical School) for allowing the use of the Patient-Practitioner Orientation Scale and for his valuable contribution to the review of the study, as well as institutional support from the Postgraduate in Health Sciences at Federal University of Goiás (UFG) and the Evangelical University of Goiás Medical School (UniEVANGÉLICA).

## Author Contributions

**Conceptualization:** Higor Chagas Cardoso, Edna Regina Silva Pereira, Viviane Soares, Marcelo Fouad Rabahi.

**Data curation:** Higor Chagas Cardoso, Edna Regina Silva Pereira, Viviane Soares, Marcelo Fouad Rabahi.

**Formal analysis:** Higor Chagas Cardoso, Edna Regina Silva Pereira, Viviane Soares, Marcelo Fouad Rabahi.

**Funding acquisition:** Higor Chagas Cardoso, Edna Regina Silva Pereira, Viviane Soares, Marcelo Fouad Rabahi.

**Investigation:** Higor Chagas Cardoso, Edna Regina Silva Pereira, Viviane Soares, Marcelo Fouad Rabahi.

**Methodology:** Higor Chagas Cardoso, Edna Regina Silva Pereira, Viviane Soares, Marcelo Fouad Rabahi.

**Project administration:** Higor Chagas Cardoso, Edna Regina Silva Pereira, Viviane Soares, Marcelo Fouad Rabahi.

**Resources:** Higor Chagas Cardoso, Edna Regina Silva Pereira, Viviane Soares, Marcelo Fouad Rabahi.

**Software:** Higor Chagas Cardoso, Edna Regina Silva Pereira, Viviane Soares, Marcelo Fouad Rabahi.

**Supervision:** Higor Chagas Cardoso, Edna Regina Silva Pereira, Viviane Soares, Marcelo Fouad Rabahi.

**Validation:** Higor Chagas Cardoso, Edna Regina Silva Pereira, Viviane Soares, Marcelo Fouad Rabahi.

**Visualization:** Higor Chagas Cardoso, Edna Regina Silva Pereira, Viviane Soares, Marcelo Fouad Rabahi.

**Writing – original draft:** Higor Chagas Cardoso, Edna Regina Silva Pereira, Viviane Soares, Marcelo Fouad Rabahi.

**Writing – review & editing:** Higor Chagas Cardoso, Edna Regina Silva Pereira, Viviane Soares, Marcelo Fouad Rabahi.

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
