## [Decision Letter · Decision Letter 0]

3 Jul 2024

PONE-D-23-44072Influence of teaching a structured and humanized method of care on the perception of medical student attitudes in the doctor-patient relationshipPLOS ONE

Dear Dr. Chagas Cardoso,

Thank you for submitting your manuscript to PLOS ONE. After careful consideration, we feel that it has merit but does not fully meet PLOS ONE’s publication criteria as it currently stands. Therefore, we invite you to submit a revised version of the manuscript that addresses the points raised during the review process. 

We look forward to receiving your revised manuscript.

Kind regards,

Delfina Fernandes Hlashwayo, Ph.D.

Academic Editor

PLOS ONE

2.  In your evaluation about this manuscript describing education research, please assess whether the authors have provided sufficient information about their teaching intervention to allow others to replicate their study, such as detailed curriculum, description of texts or methods used, or other supporting educational material. If materials, methods, and protocols are well established, authors may cite articles where those protocols are described in detail, but the submission should include sufficient information to be understood independent of these references (https://journals.plos.org/plosone/s/submission-guidelines#loc-materials-and-methods). Please do not hesitate to contact me us at plosone@plos.org if you have any questions about this submission.

Additional Editor Comments (if provided):

Reviewers' comments:

Reviewer's Responses to Questions

**Comments to the Author**

1. Is the manuscript technically sound, and do the data support the conclusions?

Reviewer #1: Yes

2. Has the statistical analysis been performed appropriately and rigorously? 

Reviewer #1: Yes

3. Have the authors made all data underlying the findings in their manuscript fully available?

Reviewer #1: Yes

4. Is the manuscript presented in an intelligible fashion and written in standard English?

Reviewer #1: Yes

5. Review Comments to the Author

Reviewer #1: Dear Authors

This study aims to assess medical students' attitudes towards doctor-patient relationships through the teaching of humanized and structured care. I commend the effort and dedication shown by the authors in this study. Among the strengths of the study are a comprehensive literature review, a clear purpose, and detailed presentation of statistical analyses.

However, to further strengthen the paper and increase its chances of publication, some suggestions can be made:

1. In the introduction section, a more detailed explanation of why humanized and structured care is important in medical education and how this study contributes to the existing literature can be provided.

2. While the methodology section provides a solid foundation, details such as how the surveys were conducted (face-to-face or online) should be elaborated. Additionally, how the initial surveys were paired with the final test could be explained. Elaborating on the methodological aspects of the study will provide a more balanced view of its strengths and weaknesses.

3. It is recommended to review the language and flow throughout the paper. Some sentences are overly long and complex. Simplifying these sentences could enhance the readability and professional appearance of the paper.

4. This indicates a significant problem with the sample size. The intervention group is only 23 participants. It is advisable to highlight this limitation in the limitations section.

5. The authors should delve into a comprehensive discussion of the results, particularly concerning the observed differences between the intervention and control groups. Providing insights into the reasons behind these discrepancies, especially in the control group, would enrich the interpretation of the findings. Additionally, addressing the divergence of results from previous studies warrants thorough exploration to contextualize the significance of their findings.

6. Based on the results of the study, suggestions for future research can be made. This may guide future studies in this field.

7. Page 13--- It has been stated that the results of this study are different from other studies. It would be appropriate for the authors to discuss the reason for this situation in detail.

8. Why is RABAHI written in capital letters on page 6?

6. PLOS authors have the option to publish the peer review history of their article (what does this mean?). If published, this will include your full peer review and any attached files.

Reviewer #1: No

---

## [Author Response · Author response to Decision Letter 0]

17 Sep 2024

Anápolis, September 17, 2024.

Dear Dr. Delfina Fernandes Hlashwayo, 

Academic Editor PLOS ONE

We thank the reviewers for their valuable comments. We have made a careful effort to comply with all the suggestions made by the reviewers and to improve the text to solve the problems noted by them. 

Below, we present a detailed point-by-point explanation of the changes made. 

We hope that the revised version is now suitable for publication.

Sincerely yours,

Authors

 Reviewer’s Comments 

We thank reviewer for the important corrections requested. We did our best to comply with all of them, and we believe they led to an improvement of the text. The changes made are described in the following paragraphs.

Comment from Reviewer, and responses:

1. In the introduction section, a more detailed explanation of why humanized and structured care is important in medical education and how this study contributes to the existing literature can be provided.

Reply: We have modified the text, and believe that the modification addresses the concerns expressed by Reviewer.

“This knowledge promoting the development of skills that will qualify the care offered by future doctors when they are in the job market, and enabling them to come closer to finding solutions to the health problems faced by the country's population.

The existing literature on medical education already demonstrates the importance of technical-scientific and biological knowledge for the training of medical professionals. However, it is necessary to expand the discussion on the benefits of teaching elements of humanization policies in the competencies of medical curriculum guidelines. Therefore, studies like the current one are needed to demonstrate how the development of these teaching strategies, that increase students' recognition of patients' expectations and autonomy, can be implemented, in addition to improving the perception of a more humanized and beneficial doctor-patient relationship [10].”

2. While the methodology section provides a solid foundation, details such as how the surveys were conducted (face-to-face or online) should be elaborated. Additionally, how the initial surveys were paired with the final test could be explained. Elaborating on the methodological aspects of the study will provide a more balanced view of its strengths and weaknesses.

Reply: We have made changes to the text.

“The PPOS scale and the sociodemographic and curricular questionnaire were applied to medical students face-to-face, randomized into two groups. Double blinding was carried out by an assistant professor from the Medical Skills department, in the fifth semester of the Medicine course, without the influence of the researchers. In this subject, students were divided into groups of 10, totaling ten groups (G1-G10). The activities were carried out in a rotating format, between the classroom (50%, or five groups) and the university pulmonology outpatient clinic (remaining groups). The teacher explained the study to the students and that they would be allocated into two groups, but would not know whether they were in the experimental or control group. Although the invitation was extended to the entire class, only 80 students agreed to participate (Fig 1) [19]. A draw was subsequently carried out to determine which groups would be subjected to the experimental protocol.”’

Figure 1. Consolidated Standards of Reporting Trials CONSORT Flow Diagram for participant recruitment and randomization

“To verify the interaction between the initial surveys and the final test and to compare PPOS scale measures pre and post-intervention, the variation in the means for each group was used, through the unpaired Student’s t-test and Mann Whitney test. In addition, the Cohen’s d effect size (ES) was calculated [20], the value considered significant for p was ≤ 0.05)’’ 

3. It is recommended to review the language and flow throughout the paper. Some sentences are overly long and complex. Simplifying these sentences could enhance the readability and professional appearance of the paper.

Reply: We thank the reviewer for the important corrections requested. We have done our best to comply with all of them, and we believe they have led to an improvement in the text. 

4. This indicates a significant problem with the sample size. The intervention group is only 23 participants. It is advisable to highlight this limitation in the limitations section.

Reply: We have made changes to the text.

’’In addition, due to the pandemic period in which the project was carried out, during the year 2020, this study had a significant problem with respect to the sample size, with the loss of 24 individuals from the study population because of medical illness. Thus, the final intervention group contained only 23 participants, restricting further evaluations between the group that received the intervention and the control group, as well as making it impossible to continue the investigation with other medical students at the study site’’ 

5. The authors should delve into a comprehensive discussion of the results, particularly concerning the observed differences between the intervention and control groups. 

Providing insights into the reasons behind these discrepancies, especially in the control group, would enrich the interpretation of the findings. Additionally, addressing the divergence of results from previous studies warrants thorough exploration to contextualize the significance of their findings.

Reply: We have modified the text, and believe that the modification addresses the concerns expressed by the Reviewer.

“Although the effects of using the SEAGULL care method proposed by Rabahi [9] have not yet been evaluated in other studies with medical students, it is suggested that its use was responsible for the differences between the intervention and control groups. 

There was a perception that learning and applying the SEAGULL care method allowed students in the intervention group to develop attitudes of empathy and patient-centered communication skills. This method of structured care provided the application of broad medical knowledge in favor of the best treatment for the patient and implementation of the attitudes of academics to center care on the patient. Therefore, it is proposed that the SEAGULL care method also provided these future doctors with the skills needed to develop and structure goals appropriate to humanized care.”

6. Based on the results of the study, suggestions for future research can be made. This may guide future studies in this field.

Reply: We have modified the text, and we believe that the modification addresses the concerns expressed by Reviewer.

“The results of the current study, which evaluated humanization in medical education, suggest the need for future research that demonstrates the importance of training programs for medical students, in order to reverse the current panorama of medical care, focused on attitudes centered on the physician and disease.”

7. Page 13--- It has been stated that the results of this study are different from other studies. It would be appropriate for the authors to discuss the reason for this situation in detail.

Reply: We have made changes to the text.

“This difference may be related to the sample size and the period of the course in which the students were enrolled, since the sample evaluated by the current study was composed of academics from the fifth academic semester of the medicine course, while other national studies analyzed academics from different undergraduate periods.”

8. Why is RABAHI written in capital letters on page 6?

Reply: We have modified the text and corrected this writing error. Marcelo Rabahi is also the author of this article and created the structured and humanized care method called SEAGULL.

---

## [Decision Letter · Decision Letter 1]

8 Nov 2024

Influence of teaching a structured and humanized method of care on the perception of medical student attitudes in the doctor-patient relationship

PONE-D-23-44072R1

Dear Dr. Chagas Cardoso,

We’re pleased to inform you that your manuscript has been judged scientifically suitable for publication and will be formally accepted for publication once it meets all outstanding technical requirements.

Kind regards,

Aima Iram Batool

Academic Editor

PLOS ONE

Additional Editor Comments (optional):

Reviewers' comments:

Reviewer's Responses to Questions

**Comments to the Author**

1. If the authors have adequately addressed your comments raised in a previous round of review and you feel that this manuscript is now acceptable for publication, you may indicate that here to bypass the “Comments to the Author” section, enter your conflict of interest statement in the “Confidential to Editor” section, and submit your "Accept" recommendation.

Reviewer #1: All comments have been addressed

2. Is the manuscript technically sound, and do the data support the conclusions?

Reviewer #1: Yes

3. Has the statistical analysis been performed appropriately and rigorously? 

Reviewer #1: Yes

4. Have the authors made all data underlying the findings in their manuscript fully available?

Reviewer #1: Yes

5. Is the manuscript presented in an intelligible fashion and written in standard English?

Reviewer #1: Yes

6. Review Comments to the Author

Reviewer #1: (No Response)

7. PLOS authors have the option to publish the peer review history of their article (what does this mean?). If published, this will include your full peer review and any attached files.

Reviewer #1: No

---

## [Editor Report · Acceptance letter]

15 Nov 2024

PONE-D-23-44072R1 

PLOS ONE

Dear Dr. Chagas Cardoso, 

I'm pleased to inform you that your manuscript has been deemed suitable for publication in PLOS ONE. Congratulations! Your manuscript is now being handed over to our production team.

Kind regards, 

on behalf of

Dr. Aima Iram Batool 

Academic Editor

PLOS ONE